# Experimental Evaluation of Graph Databases: JanusGraph, Nebula Graph, Neo4j, and TigerGraph

Jéssica Monteiro [1], Filipe Sá [1] and Jorge Bernardino [1,2,*]

1   Polytechnic of Coimbra, Coimbra Institute of Engineering (ISEC), Rua Pedro Nunes,
    3030-199 Coimbra, Portugal; a21230162@isec.pt (J.M.); filipe.sa@isec.pt (F.S.)
2   Centre for Informatics and Systems of the University of Coimbra (CISUC), Pólo II, Pinhal de Marrocos,
    3030-290 Coimbra, Portugal
*   Correspondence: jorge@isec.pt

**Abstract:** NoSQL databases were created with the primary goal of addressing the shortcomings in the efficiency of relational databases, and can be of four types: document, column, key-value, and graph databases. Graph databases can store data and relationships efficiently, and have a flexible and easy-to-understand data schema. In this paper, we perform an experimental evaluation of the four most popular graph databases: JanusGraph, Nebula Graph, Neo4j, and TigerGraph. Database performance is evaluated using the Linked Data Benchmark Council's Social Network Benchmark (LDBC SNB). In the experiments, we analyze the execution time of the queries, the loading time of the nodes and the RAM and CPU usage for each database. In our analysis, Neo4j was the graph database with the best performance across all metrics.

**Keywords:** benchmark; graph databases; LDBC SNB; NoSQL databases; open-source tools; JanusGraph; Nebula Graph; Neo4j; TigerGraph

## 1. Introduction

The development of the Internet and advances in technology have led to an increase in the size and interconnectedness of data, making it a priority to rethink the storage, connectivity, availability, security, and response time of data queries. This progress has made it possible to intensify communication between people, with the most significant technological revolution undoubtedly taking place in social networks and online platforms. The high volume of social network activity also means a significant increase in data and time spent in its management and storage in a database.

The best-known databases are relational databases, which are based on the relational data model. Although the databases that are most commonly used are relational databases, their use becomes more difficult in cases where we want to store large amounts of data. In relational databases, it is essential to avoid loss and inconsistency of data to ensure data integrity [1]. To overcome some of these shortcomings, a new type of database called NoSQL has emerged. These NoSQL databases can be of four types: document, column, key-value, and graph-oriented.

Graph databases have been increasingly studied, mainly because they are a common NoSQL database used in various domains. However, few studies have evaluated the performance and compared the features of popular graph databases.

Using the LDBC SNB benchmark, this paper evaluates the performance of the top four graph databases, as rated by the DB-Engines 2022 ranking. The DB-Engines ranking is an independent data analysis initiative that provides information on database management systems, and its main product is a monthly database popularity ranking based on several factors, including enterprise and developer adoption, online popularity, features offered, performance, scalability, community support, and feedback from expert database users. A weighted combination of these criteria is used to determine the overall DB-Engines ranking.

We selected the four top-ranked graph databases, JanusGraph, Nebula Graph, Neo4j, and TigerGraph. For each graph database, the LDBC SNB benchmark evaluates query execution time, node load time, memory, and CPU usage.

The main contributions of this work are as follows:

- Revealing the strengths and weaknesses of the top four NoSQL graph databases according to DB-Engines ranking;
- Experimental evaluation of top-ranked NoSQL graph databases using a standard benchmark;
- Identifying the best NoSQL graph database in terms of query execution time, node load time, RAM and CPU usage;
- Limitations in the practical use of NoSQL graph databases.

The remainder of this paper is structured as follows. Section 2 introduces the related work. Section 3 presents the four NoSQL graph databases. Section 4 describes the SNB benchmark of the LDBC. The experimental results are discussed in Section 5. Finally, Section 6 presents the conclusions and offers suggestions for future work.

## 2. Related Work

In this section, we present some related work dealing with graph databases and the benchmark used in this study.

The comparison of features between relational and NoSQL databases has been analyzed by Kunda and Phiri [2]. The authors compare the key features of relational and NoSQL databases to determine which type of database is better suited for modern applications and whether NoSQL could replace relational databases altogether. They provide examples of databases of each type: Memcached and Redis as key-value databases, MongoDB and CouchDB as document-oriented, Cassandra as column-oriented, and GraphDB and OrientDB as graph databases. However, one of the major weaknesses of the study is that it does not evaluate a specific type of database, let alone compare the performance of relational and NoSQL databases. The characteristics that are the bases of comparisons between the two types of databases are whether there are open-source and commercial versions, scalability, cost associated with hardware, how they handle large volumes and variety of data, availability, performance, complexity, query language, consistency, and security. Analyzing these characteristics, the authors concluded that NoSQL databases can handle large amounts of data more easily than relational databases. It is also easier to change the data schema. Although NoSQL databases do not have a standard query language, the languages used are much more accessible than SQL and can handle large amounts of data more easily.

Macak, in [3], analyzes the progress made in storing and processing large amounts of data in some graph databases where the data is highly interconnected, and there are several levels of interconnection. The authors used a Microsoft dataset and ran simple and complex queries in Neo4j and PostgreSQL, using only directly targeted data, to prove that these graph databases outperform relational databases on large volumes of data. Each query was run five times, and the highest and lowest values were taken from the results to calculate the final average. The study concluded that PostgreSQL outperformed Neo4j because the data was grouped into three sets, making it easier for PostgreSQL to access. Therefore, the experimental design may have favored one database over another. In addition, the average memory consumption of both data loading and query execution was not analyzed, with the analysis focusing only on the evaluation metric.

Fernandes and Bernardino [4] compare the main differences between relational and NoSQL databases. They also analyze the main features of the graph databases AllegroGraph, ArangoDB, InfiniteGraph, Neo4j, and OrientDB. The authors argue that for a database to be complete and practical, it must have a flexible schema, a simple query language, and good scalability. The study concludes that graph databases offer better performance and flexibility than do relational databases. It also concludes that Neo4j and ArangoDB offer good features. However, the authors did not present an experimental

evaluation of the databases, so the features mentioned were not tested and proven by testing the software of each database.

Rusu and Huang [5] analyze only two graph databases, Neo4j and TigerGraph, using the LDBC SNB benchmark. The authors perform an analysis of query execution time using scaling factors of 1, 10, 100, and 1000 on the local and cloud software provided by the databases studied. They also consider the data loading time for each scaling factor. From the experimental evaluation, the authors conclude that TigerGraph outperforms Neo4j for most queries. However, the former becomes more limited as the scale factor increases. In terms of data loading, Neo4j performs better.

The Nebula Graph database engine was studied by Wu in [6], where he presents its main database features, such as the architecture, scalability, storage, partitioning strategies, data processing, and the query language used. Several data-insertion and update tests are also performed to evaluate its performance. The performance of the database was analyzed using the LDBC SNB benchmark, using only short queries and the insert and update queries, leaving out the complex queries. It is not possible to conclude whether Graph Nebula is a graph database with excellent performance because it was not compared with other databases.

In Ref. [7], Falcão et al. compare the graph databases Neo4j, JanusGraph, and Dgraph using a version control system called Git. The study analyzed read/write operations and the performance of tasks such as inserting new data into graphs. The authors concluded that Neo4j outperformed the other graph databases, offering better read and write performance and the ability to handle high data loads.

Timón-Reina et al. in Ref. [8] analyze the evolution of Neo4j and TigerGraph databases compared to relational databases and show that it is possible to adopt graph databases in several areas, especially in biomedicine, where they have a more intuitive data visualization and representation for the user. The study concludes that graph databases offer better scalability options, more accessible query languages, and a more intuitive data representation.

Macak et al. in Ref. [9] compare multi-model and single-model databases because the multi-model combines the most advantageous features of different databases. Although a multi-model database can be versatile, it runs the risk of performing worse than single-model databases. The study compares the multi-model database OrientDB with Neo4j, a graph database, and with MongoDB, which is document-oriented. The comparison is based on the performance of the databases when performing various queries focused on multiple properties. The queries were divided into two distinct categories, wherein the first was graph queries, and the second was document-type data. For graph queries, OrientDB outperformed Neo4j at greater depths when finding the shortest path between nodes with different properties. When it came to traversing other nodes of depth-three, though, Neo4j proved the best. The document type data queries focused on the difference between an indexed and an unindexed query and concluded that MongoDB has a lower query execution time, as compared to OrientDB.

Erdemir et al. in Ref. [10] used regular graph and hypergraph models in graph databases. The performance of the Neo4j graph database was evaluated using 12 queries for each model, and the queries involved hyperedges and binary edges. The study concludes that the hyper-graph model uses more storage because additional nodes are required for this model, resulting in more associated benefits. However, the execution time is significantly reduced for queries involving multiple node types.

Hölsch et al. in Ref. [11] state that graph databases outperform relational databases when querying data. The authors note that while the back-end provides secure transactions and storage, native graph databases attract user attention by promising better performance than that of traditional databases. The study analyzes a set of analytical queries in Neo4j and two leading commercial relational databases, the identities of which were withheld from publication due to licensing restrictions. In addition to these queries, pattern-matching queries were also used, including paths that filter specific node labels without any edge-type restrictions. They also used methods to filter an edge-type without node label constraints,

including ways to filter both node and edge labels and retain paths that contain cycles. The execution times of cycle queries in Neo4j were compared with the execution times of SQL queries in the relational database. The study concludes that relational databases outperform Neo4j for analytical queries, but Neo4j is faster for queries that do not filter specific edges.

Deutsch et al. [12] present a detailed study of the main features of the graph database TigerGraph. The authors state that TigerGraph is a native parallel graph database because its storage supports nodes, edges, and node properties in a way that promotes an engine that parses queries and analysis in massively parallel processing (MPP) and is highly scalable. TigerGraph's query language, GSQL, allows users to analyze graphs in sophisticated ways. The database has iterative algorithms, such as PageRank, loosely coupled components, shortest paths, and recommender systems. GSQL supports MapReduce and its graph is implemented accordingly to support parallel processing.

Furthermore, TigerGraph has a free version and provides the necessary documentation for its use. The authors performed an experimental evaluation using a benchmark for MPP graph databases that allows the exploration of the multicore/single machine setting, analyzing data load, query execution time, and resource consumption. The study concludes that TigerGraph can handle a large volume of data with low node loading, query execution time, and reduced resource consumption. However, the study did not compare it with other databases.

Kaliyar in Ref. [13] reviews the different graph databases and applications and compares their models based on selected properties. The databases analyzed were Neo4j, DEX, InfiniteGraph, Infogrid, HyperGraphDB, Trinity, and Titan. The study proves that most graph databases have different data structures which allow query APIs and other types of query languages in each graph database. However, there was no experimental evaluation of the databases.

Lissandrini et al. in Ref. [14] present a study of existing graph databases that uses a new micro-benchmarking framework to provide insights into the performance of graph databases that goes beyond what macro-benchmarks can provide. The micro-benchmarking framework includes the most comprehensive set of queries and operators yet considered at the time of the paper's publication. The study used a large dataset to evaluate graph databases on synthetic and real data from different domains. The databases evaluated were: ArangoDB, BlazeGraph, Titan, OrientDB, Neo4j, Sparksee, and Sqlg. The study concludes that the use of microbenchmarks offers benefits to developers and researchers and can help them better understand the design, performance, and feature choices of graph databases.

This paper differs from those of the other authors presented in this section, in that we used the LDBC SNB benchmark on the latest available versions of the four graph databases with the best score in the DB-Engines 2022 Ranking.

## 3. Graph Databases

Graph databases are a type of NoSQL database in which data is represented as graphs with nodes and edges. Nodes represent entities in a database and can have associated properties and labels. A graph database is based on a graph data model of properties or labels that provide internal structures with nodes and edges [15]. This type of graph provides additional features to make the graph easier to understand, where nodes can have one or more labels, and relationships between nodes can contain properties. The edges represent relationships between different nodes, and a relationship is a link between a source node and a destination node. Graph databases are characterized by user interactivity, ease of understanding, and flexibility of the data schema. This database has gained significant notoriety in several fields, namely biology, chemistry, fraud detection, and military research, but especially in social networks. The three main characteristics of graph databases are:

- Performance: In relational databases, queries run on them become slower as the number and depth of relationships increase. In graph databases, however, performance does not change drastically as data grows.

- Flexibility: The structure and schema of a graph model adapts itself to the needs of the application, and it is possible to add new data without compromising its functionality.
- Agility: The continuous evolution of graph databases is consistent with today's agile development practices, allowing the data to evolve as users' needs change.

In the following, we describe the main characteristics of the four analyzed graph databases and present their main strengths and weaknesses.

*3.1. JanusGraph*

JanusGraph was released in 2017 and is an open-source graph database. According to Qiao [16], JanusGraph is a distributed graph database based on another graph database called Titan. It is also based on Tinker-Pop, an open-source framework for graph databases and graph analytics systems. This framework allows programmers to add graph computing capabilities to their applications without having to worry about developing APIs, graph processing engines, or algorithms. Scalability is a critical feature in graph databases. JanusGraph is scalable and can store and query large graphs distributed across a cluster of multiple machines [7]. According to the documentation available on the JanusGraph website, one can see that the database supports third-party applications for data storage and indexing. For data storage, it supports Apache Cassandra, Apache HBase, and Oracle Berkeley DB Java Edition. The indexes allow more complex queries and use Elasticsearch, Apache Solr, and Apache Lucene. It supports the Java programming language and uses the Gremlin query language. JanusGraph has the following characteristics [17]:

- Elastic and linear scalability for a growing database and number of users;
- Support for a variety of storage and indexing back-ends.

The main advantages of JanusGraph are [17]:

- High availability from multiple data centers;
- Dynamic backups.

We can also identify some limitations [17]:

- It has a variety of storage and indexing back-ends, which makes JanusGraph dependent on third parties;
- It is difficult to predict its future development.

Figure 1 represents the graph database visualization using Gremlin-Visualizer, an interactive graph visualization tool for graph databases using the Gremlin query language.

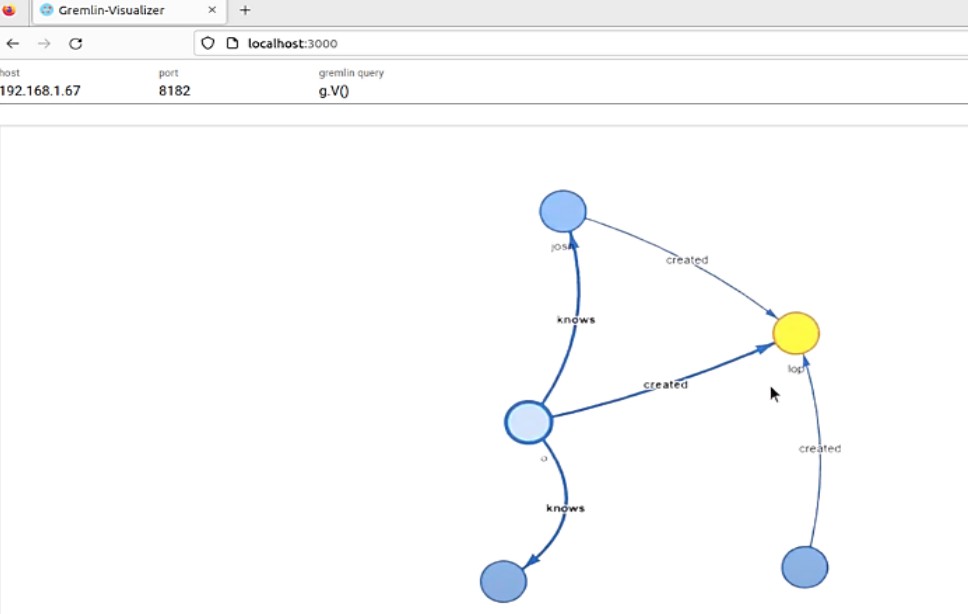

**Figure 1.** Data visualization in Gremlin-Visualizer.

### 3.2. Nebula Graph

Nebula Graph was released by Vesoft Inc. as a native database specialized in storing various graph connections and retrieving information from them. It also stores nodes and edges in graphs with properties and labels. As a query language, it uses nGQL (nebula graph query language), similar to SQL, and designed for programmers and ordinary users. Nebula Graph supports C++, Java, Python, and Go programming languages.

The main features of Nebula Graph are as follows [18]:

- Ability to host graphs with hundreds of billions of nodes and trillions of edges;
- Fast queries with a millisecond latency.

As main advantages, we can highlight the following:

- Due to its three-part architecture, it brings benefits such as high availability and cost-effectiveness by providing a higher resource utilization rate;
- As an SSD-based product, as compared to HDD and large memory products, it is more suitable for future hardware trends and easier to achieve balanced read and write.

However, it also has some limitations [18]:

- If we back up a part of a particular graph in cluster A, the backup files cannot be restored in another cluster B;
- It takes up a relatively large amount of disk space.

Nebula Graph Studio is used for data representation. This is a browser visualization tool that provides an interactive user experience. It also allows manipulation of the data schema, import of data, and execution of nGQL statements for possible retrieval. Figure 2 shows the data representation in Nebula Graph Studio.

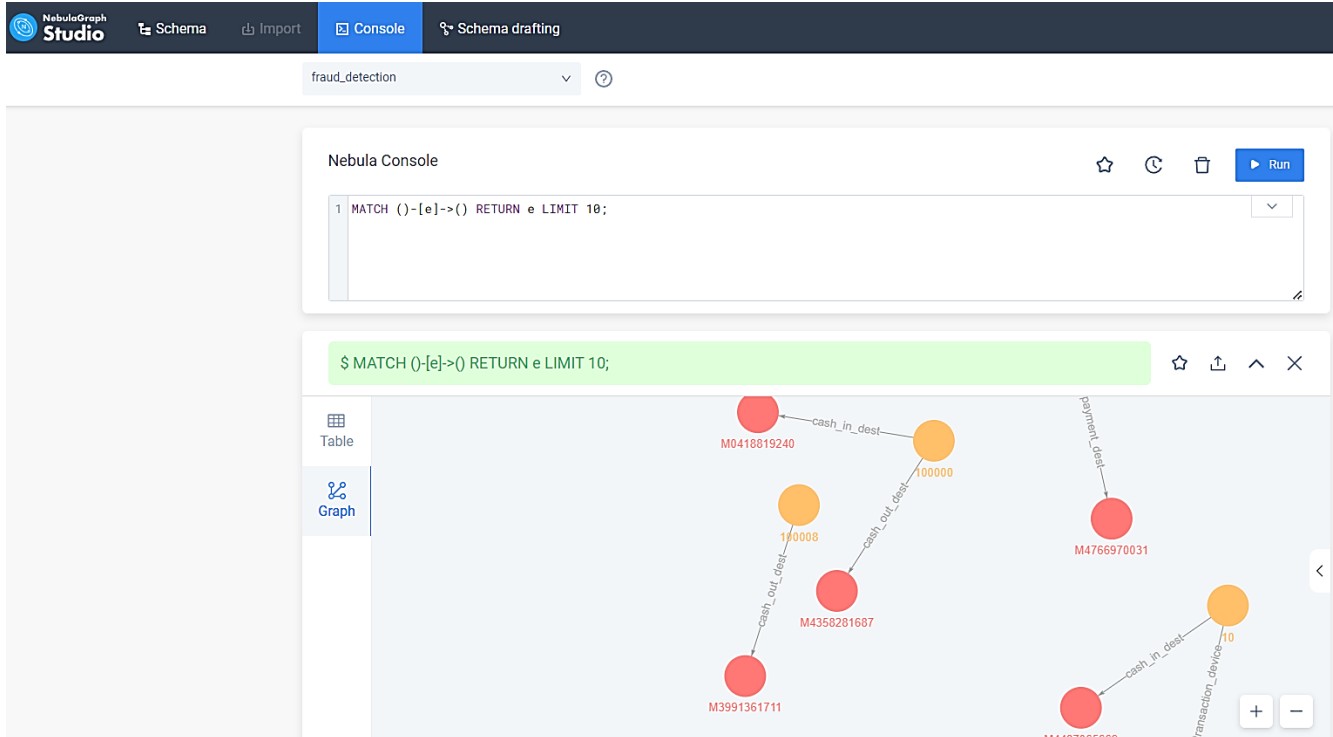

**Figure 2.** Data visualization in Nebula Graph Studio.

### 3.3. Neo4j

Neo4j is an open-source Java graph database released in 2007. It uses a persistent Java engine where it is possible to store graph structures instead of tables. It is also one of the most widely used graph databases in several domains, such as healthcare, government,

automated manufacturing, military, and others [19]. The query language is Cypher, which is inspired by SQL, and allows us to focus on the data we want from the graph [20].

In DB-Engines' 2022 ranking, we can see that Neo4j is the first-ranked database that stands out for its features [20]:

- Scalable database optimized for storing and querying large graphs distributed in a cluster on multiple machines;
- Flexible and suitable for handling data with unstructured formats;
- It has an easy-to-understand query language.

Its main advantages are [20]:

- It has a high-performance distributed cluster architecture;
- Graph fragmentation minimizes query latency;
- It has a cloud service called AuraDB and is fully managed with automatic updates and backups.

Neo4j also has some limitations, namely [20]:

- It does not directly accept RDF-formatted data;
- It consumes a large amount of memory.

The Neo4j interface is interactive, intuitive, and easy to use, as shown in Figure 3.

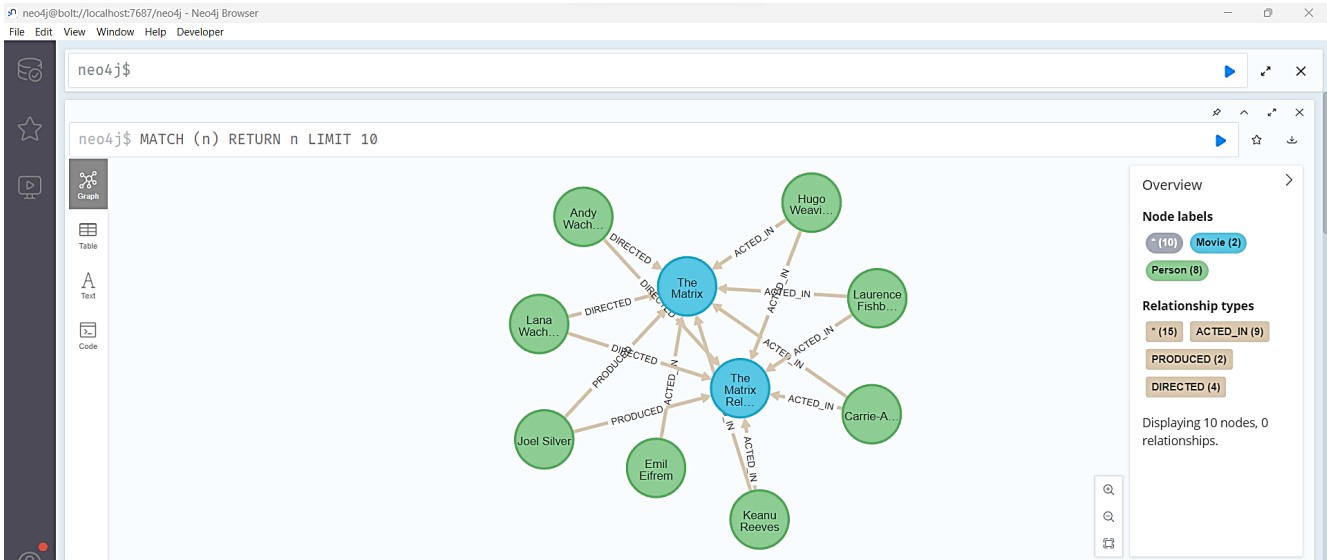

**Figure 3.** The Neo4j interface.

### 3.4. TigerGraph

TigerGraph is an open-source graph database that was released in 2017. It is a system designed to perform multiple computations simultaneously based on parallelism [12]. It is also a distributed database capable of analyzing web-scale data in real-time. Its query language is GSQL, and as the name suggests, it is a direct extension of SQL intended for graph databases, and it enforces strict declaration. TigerGraph uses C++ as its programming language.

TigerGraph has the following features [21]:

- A database capable of handling graph grids and a workload in a natural production environment where tens of terabytes of data are connected and constantly updated;
- It has a high availability cluster that uses replication to provide continuous service when one or more servers are unavailable, or some service components fail.

The main advantages of TigerGraph are as follows [21]:

- In terms of elasticity, users often do not know what hardware or computing power they will need. Elasticity eliminates the need to plan for the capacity;

- It includes a flexible, high-performance data loader that can transfer tabular or semi-structured data while online.

On the other hand, its limitations are [21]:

- It only runs on Linux servers;
- It has an expensive cloud service;
- It incurs an expensive annual subscription price for the storage it provides.

TigerGraph's interface is also very intuitive; it is shown in Figure 4.

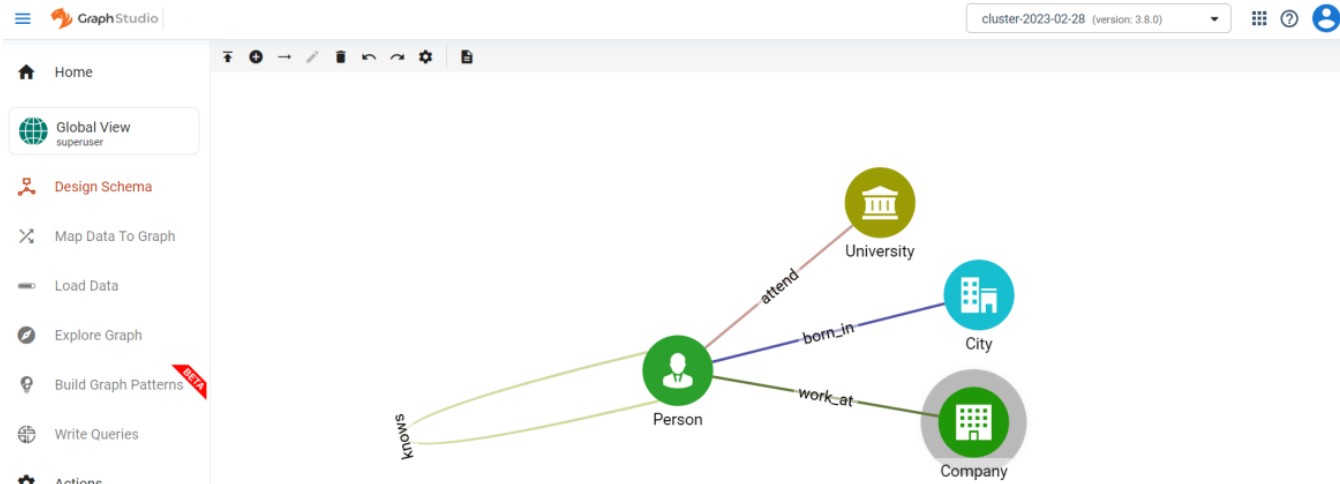

**Figure 4.** TigerGraph Studio.

## 4. LDBC SNB Benchmark

The experimental evaluation uses the Linked Data Benchmark Social Network Benchmark (LDBC SNB) from the Linked Data Benchmark Council. This is a well-known benchmark in the scientific community, and it is used to test the performance of various graph databases. Its datasets are an ideal example for graph databases because they represent real social networks where people are connected and interact with each other. This data is very relevant to graph databases because the connections between people and their interactions in social networks are naturally represented by graphs, which is the underlying data structure of graph databases. This means that the results from this benchmark are highly relevant to real-world use cases. These datasets have several scaling factors associated with their size. The larger the scale factor, the larger the number of nodes and the relationships between them. We can use this data to evaluate the behavior of databases when dealing with large amounts of data.

In addition, the LDBC SNB queries are also designed to evaluate the performance of graph databases in realistic usage scenarios, making the data even more representative and relevant to the scientific community. The benchmark includes 29 queries, some of which are more complex with many nodes, others simpler, and there are some that perform updates on the data, allowing the researcher to evaluate how graph databases handle different types of queries and data loading, as well as how they handle real-time updates.

All of these ways to evaluate graph database performance provided by the LDBC SNB allow for a fair comparison between different graph databases. The community involved in this benchmark is active and collaborative, which means that new datasets and queries are added regularly to make the LDBC SNB increasingly representative and challenging.

Figure 5 shows the graphical scheme of the benchmark. It consists of people who live in a particular city belonging to a certain country and continent, who are users who belong to a forum where it is possible to post comments, and people who know others and send messages to each other.

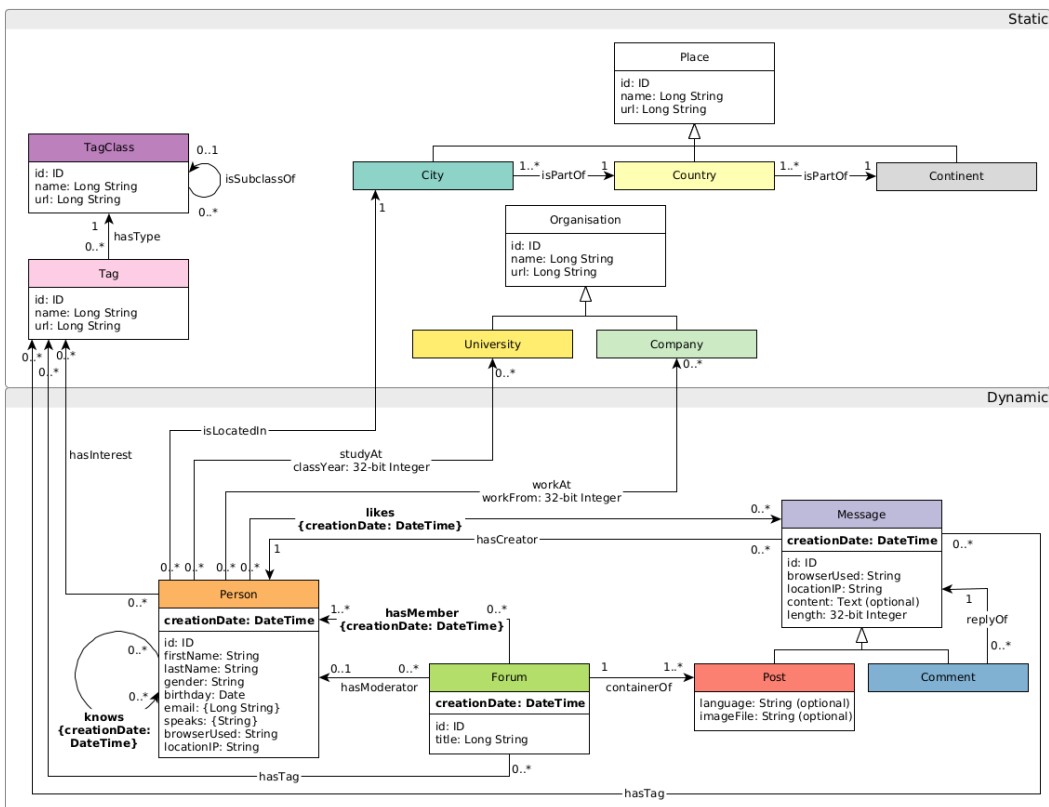

**Figure 5.** LDBC SNB graph schema (Source: [22]).

The schema is divided into two parts: Static, in which the number of nodes is maintained across all scale factors, and Dynamic, where the number of nodes and relationships increases as the scale factors also increase.

The entity with the most records is Comment which has 203,354 nodes and 1,016,770 properties at a scale factor of 0.1 and 24,271,888 nodes and 121,359,440 properties at a scale factor of 10.

*LDBC SNB Query Example*

In addition to using the benchmark queries, we also used the query languages of each graph database to import the data. Each graph database was evaluated against all 29 LDBC SNB queries.

For example, regarding the benchmark queries, Query 8 aims to find the most recent comments that respond to the person's messages on the parent node, considering only direct replies. The result of Query 8 returns the comments and the person who created each reply. Query 8 is written in Cypher, as shown in Figure 6.

```
MATCH    (start:Person    {id:    $personId})<-[:HAS_CREATOR]-(:Message)<-[:REPLY_OF]-(comment:Comment)-[:HAS_CREATOR]->(person:Person)
RETURN
person.id AS personId, person.firstName AS personFirstName,
person.lastName AS personLastName,
comment.creationDate AS commentCreationDate,
comment.id AS commentId, comment.content AS commentContent
ORDER BY
commentCreationDate DESC, commentId ASC
LIMIT 20
```

**Figure 6.** Example of a query.

## 5. Experimental Evaluation

The experimental evaluation analyzes the performance of the latest versions of the databases JanusGraph (0.6.2), Nebula Graph (3.2.0), Neo4j (1.4.15), and TigerGraph (3.0). Query execution time, node load time, and CPU and RAM consumption were analyzed using different dataset sizes. The datasets were loaded with the scale factors 0.1, 0.3, 1, 3, and 10, corresponding to the following dataset sizes:

- Scale factor 0.1: 112 MB.
- Scale factor 0.3: 339 MB.
- Scale factor 1: 1.22 GB.
- Scale factor 3: 3.68 GB.
- Scale factor 10: 12.2 GB.

### 5.1. Experimental Setup

The hardware used in this evaluation was a laptop computer with the following characteristics:

- AMD Ryzen 5 5600H CEZANNE processor, 3.3 Ghz;
- 16Gb of RAM memory;
- NVidia Ampere Geforce RTX 3060 graphics card;
- 512Gb SSD disk;
- Windows 11;
- JanusGraph version 0.6.2;
- Nebula Graph version 3.2.0;
- Neo4j version 1.4.15;
- TigerGraph version 3.0.

### 5.2. Methodology

The execution of the benchmark tests was carried out as follows:

- Installation of the graph databases;
- Loading the data from the LDBC SNB dataset and analyzing the load time;
- Running the 29 benchmark queries and extracting the execution time, as well as CPU and RAM used;
- Running the 29 queries five times each. We executed from the first query to the last and repeated the process five times to check how the engines handled the caching effect. After recording the times obtained, the result was the average of the five executions.

### 5.3. Experimental Results

The results are grouped by average node loading time, average query execution time, and average CPU and RAM consumption for query execution.

We first compared the node loading results between the scale factors in each graph database when loading nodes. The percentage differences in node loading time between the scale factors for JanusGraph, Nebula Graph, Neo4j, and TigerGraph are shown in Table 1. It can be seen that, for all graph databases, there is a more significant percentage difference when going from scale factor 0.3 to 1 and from scale factor 3 to 10. This significant difference is due to the increase in the size of the scale factor datasets. This means that the size of the datasets more than triples when moving from one scale factor to the other. It should also be noted that as the scale factor increases, the number of nodes and relationships increases, so the average time for node loading increases for prominent scale factors.

**Table 1.** Percentage increase in node load time between scaling factors for each graph database.

|  | JanusGraph | Nebula Graph | Neo4j | TigerGraph |
|---|---|---|---|---|
| SF 0.1 → 0.3 | 18% | 17% | 8% | 11% |
| SF 0.3 → 1 | 54% | 48% | 70% | 59% |
| SF 1 → 3 | 21% | 28% | 7% | 12% |
| SF 3 → 10 | 100% | 100% | 100% | 100% |

The total average time for each graph database at the different scaling factors is compared in Table 2.

**Table 2.** Average node loading time (ms) in each graph database.

|  | JanusGraph | Nebula Graph | Neo4j | TigerGraph |
|---|---|---|---|---|
| SF = 0.1 | 7044 | 9477 | 4168 | 6049 |
| SF = 0.3 | 8608 | 11,468 | 4552 | 6762 |
| SF = 1 | 18,682 | 22,181 | 15,029 | 16,494 |
| SF = 3 | 23,542 | 30,985 | 16,136 | 18,690 |
| SF = 10 | 547,642 | 609,012 | 429,412 | 486,344 |
| *Total* | *152.14 h* | *169.19 h* | *119.29 h* | *135.11 h* |

Neo4j had the lowest average node load time, along with higher scalability than the other graph databases. Although TigerGraph has very similar characteristics to Neo4j, it had the second-best average node load time, with lower scalability. The worst-performing databases were JanusGraph and Nebula Graph, with the worst average node load time, and therefore much lower scalability than Neo4j. According to the data presented in Table 2, TigerGraph had a 13% increase in average node load time compared to Neo4j, JanusGraph had a 28% increase, and Nebula Graph had a 42% increase. For each scale factor, 29 queries were executed in each graph database, grouped in three ways: fourteen complex queries, seven more specific and tapered queries called 'short,' and eight queries that performed updates.

Figure 7 shows the average execution time for each graph database (ms) for a scale factor (SF) of 0.1. Of all the executions performed, Neo4j was the database with the shortest execution time, followed by TigerGraph, then JanusGraph, and finally Nebula Graph. The query consisted of finding the shortest path between two people, and can take some time, because all paths have to be compared. All the short queries are also notable for their high execution time, as these types assume a tapered search and involve many nodes with many properties.

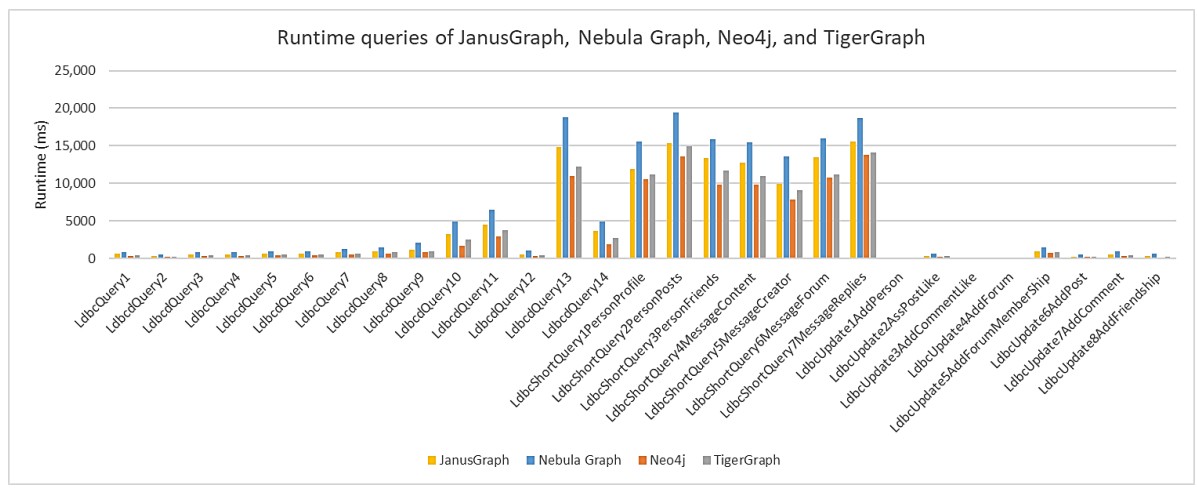

**Figure 7.** Query runtime in milliseconds for SF = 0.1.

Figure 8 shows the average query execution time (ms) in the graph databases for SF 0.3. The higher the scale factor, the larger the data size. However, the queries remain, and we again see that query 13 and the short queries have a longer average execution time for the same reasons as at a scale factor of 0.1. Nevertheless, the execution times were longer than the scale factor might indicate, because the data volume increased. The database with the shortest average execution time was Neo4j, followed by TigerGraph, JanusGraph, and Nebula Graph.

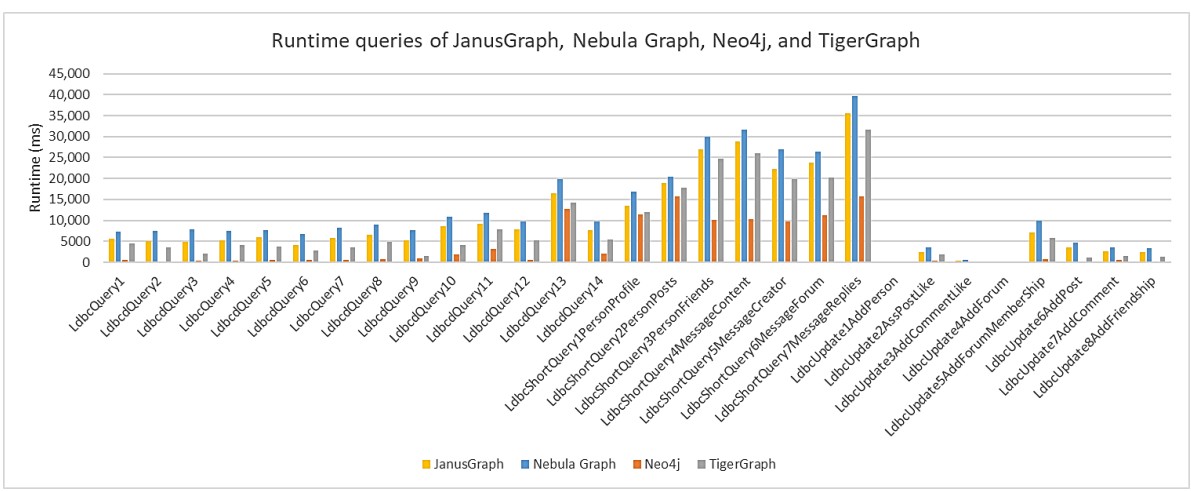

**Figure 8.** Query runtime in milliseconds for SF = 0.3.

Figure 9 also shows the average query execution time (ms) for a scale factor of 1 for each graph database. Compared to the previous scale factors, at a scale factor of 1, there is a significant increase in short queries with many nodes being the most important ones. As the scale factor increases, so does the size of the data and associated relationships. In this case, the short queries also request information about specific nodes, the number of which increases dramatically along with the scale factors. As a result, the execution time of the short queries also increases significantly. Neo4j again has the shortest execution time, followed by TigerGraph, JanusGraph, and Nebula Graph.

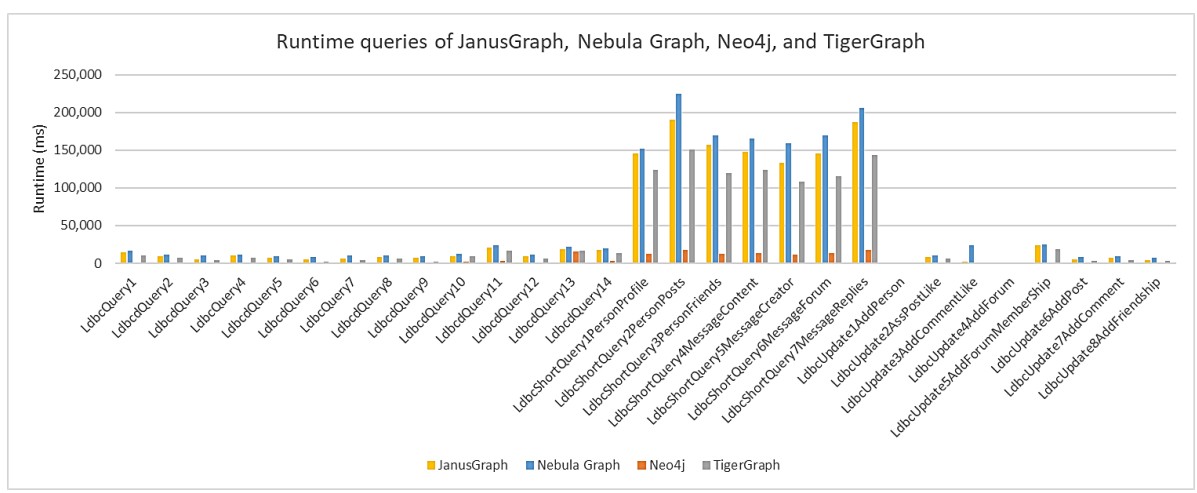

**Figure 9.** Query runtime in milliseconds for SF = 1.

Figure 10 shows the average execution time of queries in each graph database for a scale factor of 3. It is possible to see high average execution times for short queries because they request data in a larger number of nodes. These average times are higher than those shown for a scale factor of 1. For this scale factor, Neo4j is again in first place, followed by TigerGraph, JanusGraph, and Nebula Graph.

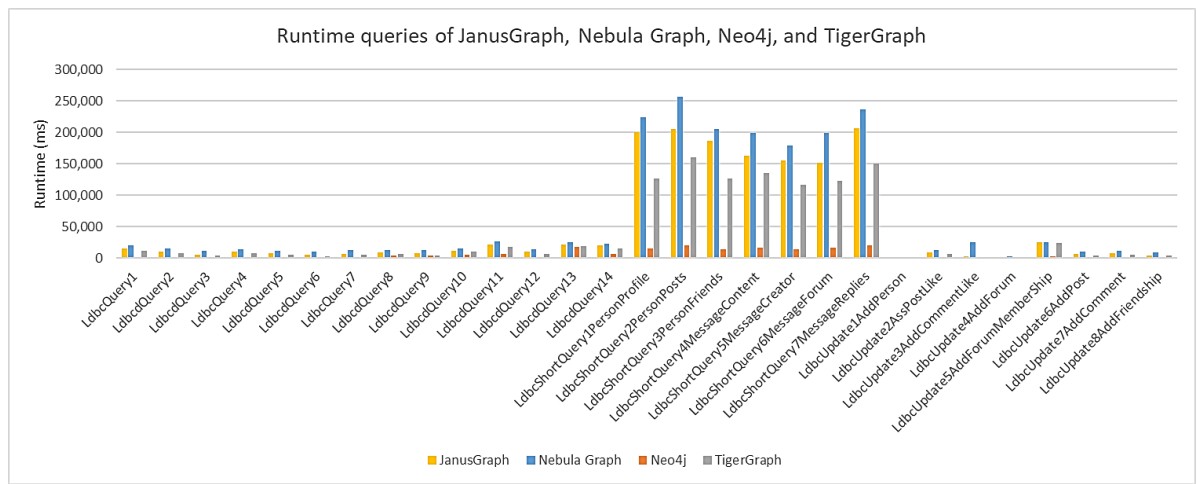

**Figure 10.** Query runtime in milliseconds for SF = 3.

Figure 11 shows the average query execution time (ms) in each graph database for SF 10. Neo4j continues to have the shortest execution time, followed by TigerGraph, JanusGraph, and Nebula Graph. At this scaling factor, the data size increased significantly from about 3 gigabytes to about 12 gigabytes. As mentioned earlier, some entities have a larger number of nodes, and the queries involving these entities require more execution time, which is the case for all short queries. Note that for short queries 1 and 2, there was a significant increase in execution time for the Nebula Graph, which is a limitation for queries involving large amounts of data.

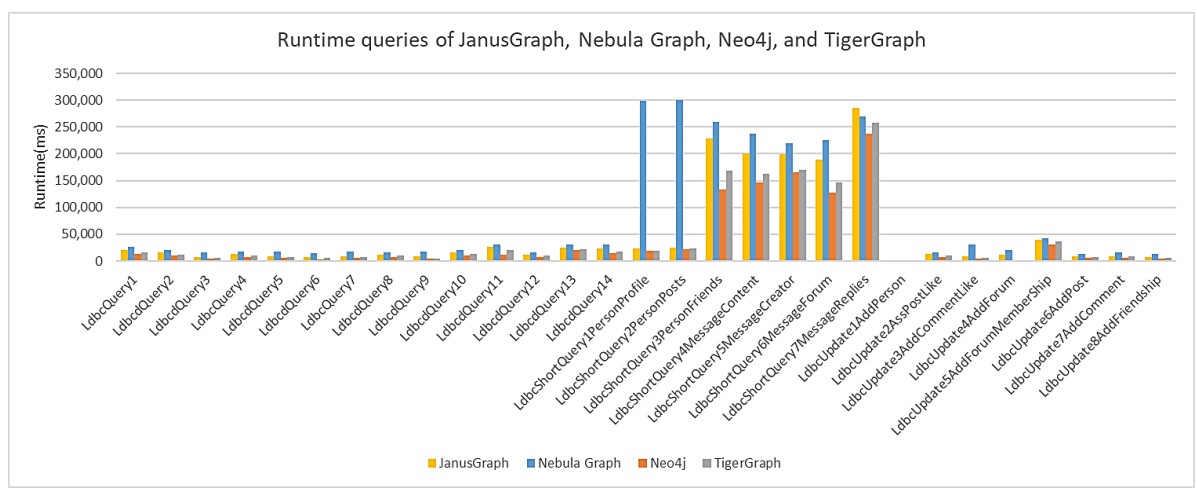

**Figure 11.** Query runtime in milliseconds for SF = 10.

The average query execution times for the four graph databases at each scale factor are shown in Table 3.

**Table 3.** Average query execution time (ms) in each graph database.

|  | Average Query Execution Time (ms) | | | |
|---|---|---|---|---|
|  | **JanusGraph** | **Nebula Graph** | **Neo4j** | **TigerGraph** |
| SF = 0.1 | 127,589.8 | 164,704.8 | 9969.2 | 111,976 |
| SF = 0.3 | 286,794.4 | 348,832.4 | 111,702.6 | 231,727.6 |
| SF = 1 | 1,307,342.2 | 1,519,484.8 | 133,962.2 | 1,029,613.0 |
| SF = 3 | 1,480,037.0 | 1,811,953.8 | 167,391.2 | 1,099,737.6 |
| SF = 10 | 1,453,228.8 | 2,249,980.2 | 1,034,978.2 | 1,185,218.2 |
| *Total* | *77.58 min* | *101.58 min* | *24.30 min* | *60.97 min* |

Across all scale factors, Neo4j stood out with a lowest average query execution time of just 24.30 min. It was followed by TigerGraph, with took more than twice as long as Neo4j. In third place was JanusGraph, with three times the time of Neo4j, and finally Nebula Graph, with more than four times the time of Neo4j.

The records of average CPU usage records for query execution are shown in Table 4. Neo4j was the database with the lowest average CPU usage, with a value of 26%, followed by TigerGraph with 37%, then JanusGraph with 41%, and Nebula Graph with 48%.

**Table 4.** Average CPU use (%) for each graph database.

|  | Average CPU Use (%) | | | |
|  | JanusGraph | Nebula Graph | Neo4j | TigerGraph |
| --- | --- | --- | --- | --- |
| SF = 0.1 | 17% | 22% | 11% | 15% |
| SF = 0.3 | 30% | 35% | 14% | 26% |
| SF = 1 | 40% | 47% | 22% | 35% |
| SF = 3 | 49% | 56% | 28% | 46% |
| SF = 10 | 69% | 78% | 57% | 63% |
| *Average* | *41%* | *48%* | *26%* | *37%* |

Figure 12 shows the execution time for all of the queries and all of the scale factors for each of the graph databases.

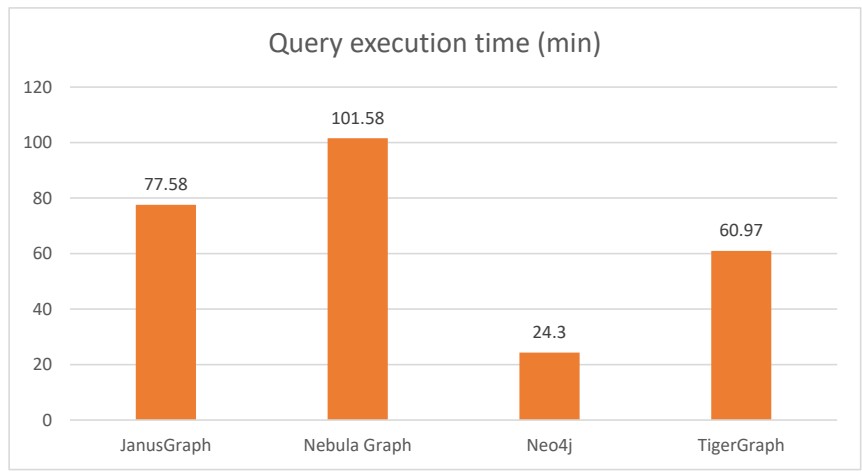

**Figure 12.** Query execution times for all queries in minutes for each graph database.

Nebula Graph consumed much more CPU than Neo4j, almost twice as much; it uses a lot of the processing power that the computer has available. This difference allows Neo4j to execute queries at all scaling factors without using all the processor's capacity.

The average RAM consumption for all queries' execution is presented in Table 5. Neo4j had the lowest average RAM consumption with 42%, followed by TigerGraph at 51%, JanusGraph at 49%, and Nebula Graph at 62%.

**Table 5.** Average RAM use (%) for each graph database.

|  | Average RAM Use (%) | | | |
|  | JanusGraph | Nebula Graph | Neo4j | TigerGraph |
| --- | --- | --- | --- | --- |
| SF = 0.1 | 33% | 39% | 29% | 31% |
| SF = 0.3 | 43% | 48% | 32% | 40% |
| SF = 1 | 54% | 62% | 37% | 50% |
| SF = 3 | 61% | 70% | 43% | 57% |
| SF = 10 | 83% | 89% | 67% | 76% |
| *Average* | *49%* | *62%* | *42%* | *51%* |

Neo4j excels at using less RAM. When we ran queries, Nebula Graph had to use more RAM in its internal data reading processes.

*5.4. Summary*

When analyzing all the tables resulting from node load, query execution, and CPU and RAM consumption, it is worth noting that Neo4j stood out from the other databases in all tests, always achieving the best results. Without a doubt, Neo4j is a graph database with specific and advanced features that allow it to efficiently handle large volumes of data. It was the database that always stood out. TigerGraph was the second-best database. Although its average RAM consumption was slightly better than that of JanusGraph, it outperformed JanusGraph in the other aspects analyzed. Despite having all the features that a graph database should have, TigerGraph still has limitations when dealing with larger datasets. Not only does it consume more memory than Neo4j, it also has a higher query execution time. JanusGraph and Nebula Graph took a long time to load data into all the datasets. Most of these loads took days to be complete, and the memory consumption for both loads and queries was very high. Both graph databases are still under development and may need to show better performance to be viable.

## 6. Conclusions and Future Work

The main objective of this study was to examine and evaluate the latest versions of the most popular graph-type databases. We selected the JanusGraph, Nebula Graph, Neo4j, and TigerGraph engines as the top four databases in the DB-Engines 2022 Ranking. The LDBC SNB benchmark was used to evaluate the performance of the four databases. From the experiments, we conclude that Neo4j outperformed the other graph databases overall in terms of node load time and query execution time. It is a database that can handle a large volume of data and guarantees high scalability and low latency.

Furthermore, it also offers better guarantees, is easy to use and compress, and has the Cypher query language, which is considered the most powerful and intuitive by users and programmers, although some prior knowledge is required. Neo4j also has a flexible data schema that allows users to easily access the data. Although TigerGraph has similar features to Neo4j, it is slower when dealing with larger amounts of data. JanusGraph and Nebula Graph are very different, as they are still under development. The applications of these databases are many and varied. We can use them for personal or experimental purposes. They are used in tax and financial fraud detection, in biochemistry and biology to identify drug components, and most importantly, in social networking.

As to future work, we intend to evaluate other open-source graph databases using the LDBC SNB benchmark, compare them to other open-source graph databases and use higher scaling factors.

**Author Contributions:** Conceptualization, J.B. and F.S.; Methodology, J.M. and J.B.; Software, J.M.; Validation, J.M., F.S. and J.B.; Formal analysis, J.M., F.S. and J.B.; Investigation, J.M.; Resources, J.M.; Data curation, J.M.; Writing—original draft preparation, J.M.; Writing—review and editing, J.B. and F.S.; Supervision, J.B. and F.S.; Project administration, J.B. and F.S.; Funding acquisition, J.B. All authors have read and agreed to the published version of the manuscript.

**Funding:** This research received no external funding.

**Data Availability Statement:** Data sharing is not applicable to this article.

**Conflicts of Interest:** The authors declare no conflict of interest.

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
