# Peer review of "Experimental Evaluation of Graph Databases: JanusGraph, Nebula Graph, Neo4j, and TigerGraph"

_applsci, doi:10.3390/app13095770_

Round 1

Reviewer 1 Report

The paper is quite good, providing a comparative study by the means of a performance benchmark of 4 graph oriented databases.

The results and the methodology are very good.

The paper has only 2 drawbacks

- it is not very well justified why only these 4 databases were used. They state the reasons in the introduction, but I am not sure they are enough.

- it is not explicitly justified why the LNBC benchmark is relevant, except from its large utilization. Does this benchmark is representative in the kind of queries? Maybe a section better explaining the benchmark would help.

Author Response

The paper is quite good, providing a comparative study by the means of a performance benchmark of 4 graph oriented databases. The results and the methodology are very good.

Response: Thank you for your kind comments.

Point 1: The paper has only 2 drawbacks.

  • It is not very well justified why only these 4 databases were used. They state the reasons in the introduction, but I am not sure they are enough.

Response 1: Thank you for your suggestion. The graph databases used have been selected because they are in the top four of the 2022 DB-Engines ranking. You can view the current Graph Databases ranking at the following link: https://db-engines.com/en/ranking/graph+dbms.

Therefore, in order to clarify this point, we have modified Section 1 and have also added the main contributions of this work by including the following text:

“1. Introduction

Using the LDBC SNB benchmark, this paper evaluates the performance of the top four graph databases, according to the DB-Engines 2022 ranking. DB-Engines ranking is an independent data analysis initiative that provides information on database management systems, and its main product is a monthly database popularity ranking based on several factors including enterprise and developer adoption, online popularity, features offered, performance, scalability, community support, and feedback from expert database users. Therefore, a weighted combination of these criteria is used to determine the overall DB-Engines ranking.

We selected the four top-ranked graph databases, JanusGraph, Nebula Graph, Neo4j, and TigerGraph. For each graph database, the LDBC SNB benchmark evaluates query execution time, node load time, memory and CPU usage.

The main contributions of this work are as follows:

  • Revealing the strengths and weaknesses of the top four NoSQL graph databases, according to DB-Engines ranking;
  • Experimental evaluation of top-ranked NoSQL graph databases using a standard benchmark;
  • Best NoSQL graph database in terms of query execution time, node load time, RAM and CPU usage;
  • Limitations in the practical use of NoSQL graph databases.

….”

Point 2: It is not explicitly justified why the LNBC benchmark is relevant, except from its large utilization. Does this benchmark is representative in the kind of queries? Maybe a section better explaining the benchmark would help.

Response: Thank you for your suggestion. The name of the benchmark is LDBC (Linked Data Benchmark Council) and it is widely used in the scientific community, in academic research, and even in industry to evaluate the performance of linked data systems and to guide the development of new systems and tools.

Therefore, to clarify this point, we have amended section 4 to include the following text:

“The experimental evaluation uses the Linked Data Benchmark Social Network Benchmark (LDBC SNB) from the Linked Data Benchmark Council. This is a well-known benchmark in the scientific community and is used to test the performance of various graph databases. Its datasets are an ideal example for graph databases because they represent real social networks where people are connected and interact with each other. This data is very relevant to graph databases because the connections between people and their interactions in social networks are naturally represented by graphs, which is the underlying data structure of graph databases. This means that the results from this benchmark are highly relevant to real-world use cases. These datasets have several scaling factors associated with their size. The larger the scale factor, the larger the number of nodes and the relationships between them. We can use this data to evaluate the be-havior of databases when dealing with large amounts of data.

In addition, the LDBC SNB queries are also designed to evaluate the performance of graph databases in realistic usage scenarios, making the data even more representative and relevant to the scientific community. The benchmark includes 29 queries, some more complex with many nodes, others simpler, and some that perform updates on the data, allowing you to evaluate how graph databases handle different types of queries and data loading, as well as how they handle real-time updates.

All of these ways to evaluate graph database performance provided by the LDBC SNB allow for a fair comparison between different graph databases. The community involved in this benchmark is active and collaborative, which means that new datasets and queries are added regularly to make the LDBC SNB increasingly representative and challenging.

Figure 5 shows the graphical scheme of the benchmark. It consists of people who live in a particular city belonging to a country and continent, users who belong to a forum where it is possible to post comments, and people who know others and send messages to each other.”

We are very grateful to the reviewer for his very helpful and detailed comments, which have contributed to the improvement of our paper.

Reviewer 2 Report

This paper presents a good comparison among different NoSQL graph databases, taking into account different parameters. I recommend to amend the following issues:

* Section "5.3 Experimental Results" where it is explained about the average node load time, and it is referring to the Table 1 (line 413), I am wondering if it has to be Table 2 instead.

* Have a look to the position of Figures 8, Figure 11 and Table 4 between the paragraphs.

Author Response

This paper presents a good comparison among different NoSQL graph databases, considering different parameters.

Response: Thanks for your kind comment.

Point 1: Section "5.3 Experimental Results" where it is explained about the average node load time, and it is referring to the Table 1 (line 413), I am wondering if it has to be Table 2 instead.

Response 1: In Section "5.3 Experimental Results", on line 413, we have corrected the numbering of the table so that it should read Table 1 instead of Table 2:

“According to the data presented in Table 2, TigerGraph had a 13% increase in average node load time compared to Neo4j, JanusGraph had a 28% increase and Nebula Graph had a 42% increase. For each scale factor, 29 queries were executed in each graph database, grouped in three ways: fourteen complex queries, seven more specific and tapered queries called short, and eight queries that perform updates.”

Point 2: Have a look to the position of Figures 8, Figure 11 and Table 4 between the paragraphs.

Response 2: Thank you for your response. The positions of Figures 8 and 11 and Table 4 have been corrected in Section "5.3 Experimental Results".

We have corrected some formatting errors in the images and tables. We have also added a new Figure 12 - Query execution time for all queries in minutes for each graph database;

We are very grateful to the reviewer for his very helpful and detailed comments, which have contributed to the improvement of our paper.

Reviewer 3 Report

Please choose a different data representation for figures 7-11. It is not clear which database is best. Maybe a linear representation would be better for this... or a linear chart of the average values on top of the bar chart.

Conclusions might be better supported by the results achieved. One sentence which stands that the Neo4j database is the best seems not to be enough. It is a place for results comparison.

English language and style required editing.

Author Response

Please choose a different data representation for figures 7-11. It is not clear which database is best. Maybe a linear representation would be better for this... or a linear chart of the average values on top of the bar chart.

Conclusions might be better supported by the results achieved. One sentence which stands that the Neo4j database is the best seems not to be enough. It is a place for results comparison.

English language and style required editing.

Response: Thank you for your suggestions.

Point 1: Please choose a different data representation for figures 7-11. It is not clear which database is best. Maybe a linear representation would be better for this... or a linear chart of the average values on top of the bar chart.

Response 1: Thank you for your comments. We agree that the linear scale helps the reader to get a better sense of the values for each bar in the graph. Therefore, in section "5.3 Experimental Results", Figures 7 to 11 now have a linear scale.

We have corrected formatting errors in the figures and tables. We have also added a new Figure 12 - Query execution time for all queries in minutes for each graph database.

Point 2: Conclusions might be better supported by the results achieved. One sentence which stands that the Neo4j database is the best seems not to be enough. It is a place for results comparison.

Response 2: Thanks for your suggestion. The presented study justifies the results section by section. The results first show that the Neo4j database is the fastest engine in terms of queries: “Across all scale factors, Neo4j stood out with the lowest average query execution time of just 24.30 minutes. It was followed by TigerGraph, with took more than twice as long as Neo4j. In third place was JanusGraph, with three times the time of Neo4j, and finally Nebula Graph, with more than four times the time of Neo4j.”

The next section analyzes CPU usage, showing Neo4j as the engine with the lowest usage: “The records of average CPU usage records for query execution are shown in Table 4. Neo4j is the database with the lowest average CPU usage, with a value of 26%, followed by TigerGraph with 37%, then JanusGraph with 41% and Nebula Graph with 48%.”

The RAM memory consumption study is also presented, “The average RAM consumption for queries execution is presented in Table 6. Neo4j has the lowest average RAM consumption with 42%, followed by TigerGraph at 51%, JanusGraph at 49%, and Nebula Graph at 62%.”

Therefore, we have presented the reasons why the Neo4j database is the best, according to the points studied and analyzed in the paper.

Point 3: English language and style required editing.

Response 3: Thank you for your comment. There has been a lot of re-writing of sentences throughout the article for the correction and improvement of all English language.

We are very grateful to the reviewer for his very helpful and detailed comments, which have contributed to the improvement of our paper.

Reviewer 4 Report

This paper studies and evaluates the latest versions of the most popular graph-type databases. The authors use the LDBC SNB benchmark to evaluate the performance of JanusGraph, Neobula Graph, Neo4j and TigerGraph. Finally, they conclude that Neo4j has the best performance.

However, this paper has no originality and novelty. As a research paper, it has no personal idea and novel methods. It just summaries other reserchers' work, and it is more like an experiment report. Moreover, this paper introduces the limitations and advantages of existing graph databases. However, the authors have no idea on solving these limitations. This paper has no academic value for the researchers. The authors must carefully consider the novelty of this paper. A paper should have its novel idea on optimizing the limitations of the state-of-the-arts solutions. For example, the authors can propose their solutions on improving the limiations of existing graph databases, and make experiments compared with the state-of-the-art graph databases to prove that their solution is better than existing ones.

Therefore, I could hardly accept the manuscript.

Author Response

This paper studies and evaluates the latest versions of the most popular graph-type databases. The authors use the LDBC SNB benchmark to evaluate the performance of JanusGraph, Nebula Graph, Neo4j and TigerGraph. Finally, they conclude that Neo4j has the best performance.

Point 1: However, this paper has no originality and novelty. As a research paper, it has no personal idea and novel methods. It just summaries other researcher’s' work, and it is more like an experiment report. Moreover, this paper introduces the limitations and advantages of existing graph databases. However, the authors have no idea on solving these limitations. This paper has no academic value for the researchers. The authors must carefully consider the novelty of this paper. A paper should have its novel idea on optimizing the limitations of the state-of-the-arts solutions. For example, the authors can propose their solutions on improving the limitations of existing graph databases, and make experiments compared with the state-of-the-art graph databases to prove that their solution is better than existing ones. Therefore, I could hardly accept the manuscript.

Response 1: Thank you for your time and comments on our paper submitted to the journal. In response to your comments, we would like to clarify a few points. Regarding the originality of our work, it is neither pure research nor the application of new methods.

Therefore, in order to clarify this point, we have modified Section 1 and have also added the main contributions of this work by including the following text:

“The main contributions of this work are as follows:

  • Revealing the strengths and weaknesses of the top four NoSQL graph databases, according to DB-Engines ranking;
  • Experimental evaluation of top-ranked NoSQL graph databases using a standard benchmark;
  • Best NoSQL graph database in terms of query execution time, node load time, RAM and CPU usage;
  • Limitations in the practical use of NoSQL graph databases.

To the best of our knowledge, this article is one of the first to apply the LDBC SNB benchmark to evaluate graph databases with the best score in the DB-Engines 2022 ranking. However, our aim was to evaluate the most popular and advanced graph databases and use a benchmark to compare their performance, in particular the execution time, node load time, RAM and CPU usage of JanusGraph, Nebula Graph, Neo4j and TigerGraph. This benchmark is based on a schema that represents a social network where the proliferation of data is visible every day and whose data is representative of the real world. Since graph databases are known for their scalability and ability to handle large amounts of data, this benchmark is perfect for this type of database. We believe that this benchmarking model is important for the scientific community of graph database researchers, as well as for companies and users to learn about the characteristics of databases and how best to adapt them to their environment. For this reason, it was not enough to simply record node load times and query execution times. It was necessary to record the performance in relation to the hardware in order to make the reader aware of the limitations and minimum requirements for a machine to support these databases when they are subjected to data loading and queries with different sizes of data sets. Regarding the limitations of existing graph databases, we agree that it is important to discuss them and explore solutions to improve them. However, our goal in this work was to evaluate and compare the graph databases, even with their limitations, to prove their strengths, not to propose innovative solutions.

In Section 2, "Related Work", we listed 14 papers resulting from research on graph databases and the LDBC benchmark. Many of them refer only to the best-known graph databases, Neo4j and TigerGraph, or even compare them with multi-model databases, but with the emergence of others, it is necessary and useful to check if they have new technologies and if they also adapt to a large amount of data, among other characteristics of graph databases mentioned in the paper. We believe that our work has scientific value and can contribute to the graph database research community.

We are very grateful to the reviewer for his very helpful and detailed comments, which have contributed to the improvement of our paper.